

**Advance prediction of coastal groundwater levels with temporal**
**convolutional network**
Xiaoying Zhang[a,b], Fan Dong[a,b], Guangquan Chen[c], Zhenxue Dai[a,b]*
[a] Institute of Intelligent Simulation and Early Warning for Subsurface Environment,
Jilin University, Changchun, China
[b] College of Construction Engineering, Jilin University, Changchun , China
[c] Key Laboratory of Marine Sedimentology and Environmental Geology, First
Institute of Oceanography, State Oceanic Administration, Qingdao, China
* Correspondence: Zhenxue Dai, dzx@jlu.edu.cn
**Highlights**
• A TCN-based model was proposed to predict groundwater levels in coastal
aquifers
• Tidal, precipitation and groundwater levels were utilized as input data in the
networks
• In advance 1- day, 3-, 7- and 15-days groundwater levels were predicted with the
highest accuracy of 1 day-lead prediction
• The TCN-based model outperforms the LSTM in accuracy and efficiency in a
coastal aquifer
**Abstract**
Prediction of groundwater level is of immense importance and challenges for the
coastal aquifer management with rapidly increasing climatic change. With the
development of artificial intelligence, the data driven models have been widely
adopted in predicting hydrological processes. However, due to the limitation of
network framework and construction, they are mostly adopted to produce only
one-time step in advance. Here, a TCN-based model is developed to predict
groundwater level variations with different leading periods in a coastal aquifer. The


historical precipitation and tidal level data are incorporated as input data. The first
hourly-monitored ten-month data were used for model training and testing, and the
data of the following three months were predicted with 24, 72, 18 and 360 time steps
in advance. For one-step prediction of the two wells, the calculated $R^2$ are higher than
0.999 in the prediction stage. The performance is meanwhile compared with a
powerful network in the field of time-series prediction, long short-term memory
(LSTM) recurrent network. The corresponding $R^2$ of the LSTM-based model are
0.996 and 0.998. While the RMSE values of TCN-based model are less than that of
LSTM-based model with shorter running times. For the advanced prediction, the
model accuracy greatly decreases with the increase of advancing period from 1-day to
3-, 7- and 15-days. Overall, the TCN- and LSTM-based models show great ability to
learn complex patterns in advance using historical data within the time series.
Considering the simulation accuracy and efficiency, the TCN-based model
outperforms the LSTM-based model and has been proved to be a valid localized
groundwater prediction tool in the subsurface environment.

**Keywords:** prediction; Groundwater level; Coastal aquifer; Temporal convolutional
networks; Long Short-Term Memory





# 1 Introduction

As the economic development and population escalate in coastal area, the fresh groundwater needs continue to mount, seawater intrusion has post great threat to the availability of portable water resources globally (Baena-Ruiz et al., 2018). In United States, Mexico, Canada, Australia, India, South Korea, Italy and Greece with dense population, numerous coastal aquifers have experienced salinization caused by seawater intrusion (Barlow and Reichard, 2009; Park et al., 2011; Pratheepa et al., 2015). Protection projects such as aquifer replenishment can be constructed to alleviate seawater intrusion by artificially increasing groundwater recharge in the aquifer than what occurs naturally (Abdalla and Al-Rawahi, 2012; Lu et al., 2019). The replenishment programs have been operated in developed area such as Perth, Western Australia, and California, USA (Garza-Díaz et al., 2019). The infrastructure tends to be costly and out of reach for many developing countries. A reliable seawater intrusion monitoring and predicting system with wells is essential and still the most effective method of keeping water chemistry above the seawater interface (Xu and Hu, 2017).

In the past several decades, conventional numerical models have been widely utilized to simulate and predict the groundwater fluctuation dynamics and chemical variations (Batelaan et al., 2003; Dai et al., 2020; Huang et al., 2015; Li et al., 2002). However, the difficulty of acquiring extensive hydrological and geological data and setting reasonable boundaries limits its application on seawater intrusion management. Meanwhile, the method is not suitable to simultaneously adopt updated monitoring





data and produce real-time prediction. Under such circumstances, where data source
is scarce, artificial intelligence technology has been proposed in groundwater dynamic
prediction. Artificial neutral network (ANN) has been greatly improved and became a
robust tool for dealing groundwater problems, where the flow is nonlinear and highly
dynamic in nature (Maier and Dandy, 2000). The conventional network model
generally has defects such as high computational complexity, slow training speed, and
failure in retaining historical information, thus is hardly to be enrolled in the
long-term time-series prediction (Cannas et al., 2006; Mei et al., 2017). To solve this
problem, researchers upgraded the conventional networks by integrating them with
methods like genetic algorithm (Danandeh Mehr and Nourani, 2017; Ketabchi and
Ataie-Ashtiani, 2015), singular spectrum (Sahoo et al., 2017), and wavelet transform
(Gorgij et al., 2017; Seo et al., 2015; Zhang et al., 2019). Singular spectrum analysis
and wavelet transform can help to preprocess the time-series data before they are put
into the neural networks to improve prediction accuracy and efficiency.

With the computing capacity development, deep learning (DL) has emerged as a

very powerful time-series prediction method. DL models are particularly suitable for
big data time-series, because they can automatically extract complex patterns without
feature extraction preprocessing steps (Torres et al., 2019). However, the general fully
connected networks are not effective to capture the temporal dependence of
time-series (Senthil Kumar et al., 2005). Therefore, more specialized DL models, such
as recurrent neural networks (RNN) (Rumelhart et al., 1986) and convolutional neural
networks (CNN) (Lecun et al., 1998) have been adopted in the field of time-series


prediction (Feng et al., 2020). Different from the back-propagation (BP) neural
network, the RNN preserves the information from the previous step as input to the
current step with loops (Coulibaly et al., 2001). This allows the RNN to handle
time-series and other sequential data but generally is not straightforward for a
long-term calculation in practice (Bengio et al., 1994). Therefore, the enhanced RNN
model, long short-term memory (LSTM) is proposed and capable to process high
variable-length sequences even with millions of data points (Fischer and Krauss, 2018;
Kratzert et al., 2019) . As one of the best deep neural network model in time-series
predicting, the LSTM has been widely used in the prediction of temporal variations
such as stock market predictions (Fischer and Krauss, 2018), rainfall-runoff (Kumar
Dubey et al., 2021) and groundwater level (Solgi et al., 2021). Despite of substantial
progresses in hydrology predicting, these networks still have issues of low training
efficiency and low accuracy (Zhan et al., 2022).

More recently, a variant of the CNN architecture known as temporal

convolutional networks (TCN) has acquired popularity (Bai et al., 2018). The
prominent characteristic of TCN is its ability to capture long-term dependencies
without information loss (Cao et al., 2021). Meanwhile, it joints a residual block
structure to fix the disappearance of gradient in the deep network structure (Chen et
al., 2020). With proper modifications, the TCN is quite genetic and easily to be used
to build a very deep and extensive network in sequence modeling. In earth science,
the TCN has been successfully applied to time-series prediction tasks including
multivariate time-series predicting for meteorological data (Wan et al., 2019),



probabilistic predicting (Chen et al., 2020) and wind speed predicting (Gan et al.,
2021). Researches suggest that the TCN convincingly has advantage in several
popular deep learning models across a broad range of sequence modeling tasks
(Borovykh et al., 2019; Chen et al., 2020; Wan et al., 2019). However, the potential of
TCN has not been investigated in the sequencing model of hydrogeology field.

Another import subject is that these networks are mostly used to predict

variables in only one step, which is not enough for the prediction of hydrology
information in management. Therefore, it is worthy to explore their prediction
abilities in longer periods. The objective of this study is to build a climate-dydro
hybrid data-driven model with TCN to develop a real-time advance prediction model
of groundwater level in the coastal aquifers. The hourly processed tidal, precipitation
with groundwater level data in monitoring wells of Laizhou Bay are utilized to train
model and prediction the groundwater level in a period of 1-day, 3-,7- and 15-days.
To further validate the accuracy and efficiency of the proposed model, its
performance is further compared with the LSTM-based model. The rest of the paper is
organized as follows. Sect. 2 introduces the study area and observational data. Sect. 3
illustrates the detailed concept of TCN and LSTM, the experimental model settings
and model evaluation criteria. Sect. 4 presents the predicting results and discussions.
Finally, the paper is concluded in Sect. 5.
**2 Study area and data processing**
**2.1 Site description**

The study area is located in the south coast of Laizhou Bay, along the Yangzi to



Weifang section in Shandong province of China (Fig. 1). The Laizhou Bay is one of
the earliest and most seriously affected area by seawater intrusion since 1970s in
China (Han et al., 2014; Zeng et al., 2016). The area is basically a coastal plain, which
contains a series of Cretaceous to modern sediments that covering the Paleozoic
basement. The sedimentary facies of coastal aquifer are alluvium, proluvial and
marine sediments from south to north (Han et al., 2011). According to the research of
(Xue et al., 2000), there have been three seawater intrusion and regression events in
the sea area of Laizhou Bay since the upper Pleistocene. The transgression in the early
upper Pleistocene formed the third marine aquifer containing sedimentary water.
These brine were formed by evaporation and concentration of ancient seawater and
re-dissolution and mixing of salt (Dai and Samper, 2006; Zhang et al., 2017). The
monitoring wells BH01-BH05 are distributed in the study area along a cross section
perpendicular to the coastline. Among the wells, the data of well BH01 and BH05 are
relatively integrate and distributed in the two sides of the cross profile with
distinguished annual variation pattern, which are selected as examples for the
developed models.

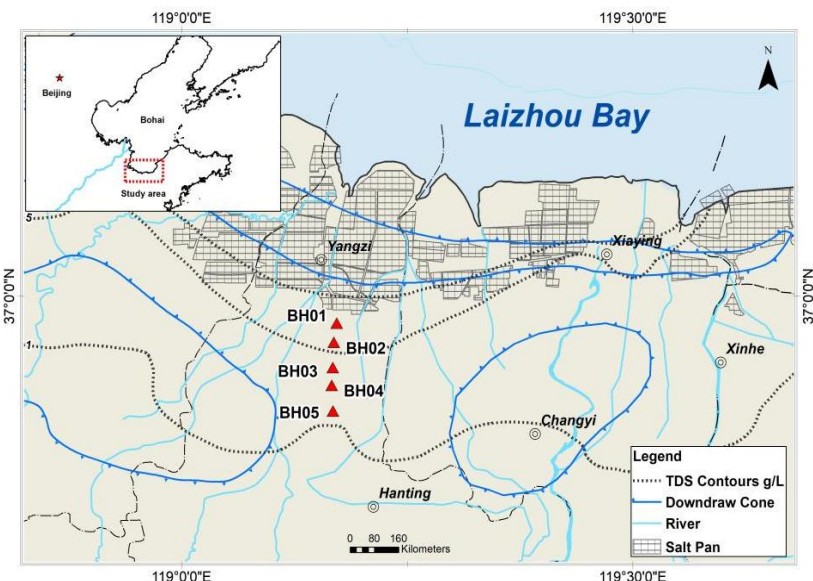

Figure 1. Schematic figure of the study area with monitoring wells BH01-BH05.

## 2.2 Data collection and preprocessing

The precipitation and tidal level are selected as the primary factors to affect the groundwater dynamics in the coastal area. The data in the period of 2011 to 2012 with groundwater level observations of three wells are combined as the input of the deep learning models. A total of 37,920 data items are collected for monitoring wells and the variations of groundwater level, and tidal level with precipitation are shown in Figure 2. The rainfall is concentrated from June to September and in shortage from December to April. The tide in the study area is irregular mixed with a semi-diurnal variation. In the experiments, ten months of data from October to July 2011 is first extracted for model training and testing. The rest of the data from August 2012 to October 2012 is used to test model prediction accuracy.

In addition, the magnitudes of meteorological and hydrological variables have





obvious temporal variations. To reduce the negative impact on the model learning
ability, especially on the speed of gradient descent, all variables are normalized to
ensure that they remain at the same scale (Kratzert et al., 2019). This preprocessing
method ensures the stable convergence of parameters in the developed TCN- and
LSTM-based models and improve the simulation accuracy of the model. The
normalization formula is as follows:
$$y_i = \frac{x_i - x_{min}}{x_{max} - x_{min}} \qquad (1)$$

where $x_i$ represents the data in time $i$; $x_{max}$ and $x_{min}$ are the maximum and minimum
variable values. The output of the network is retransformed to obtain the final
groundwater level prediction, which is an inverse data scaling process.

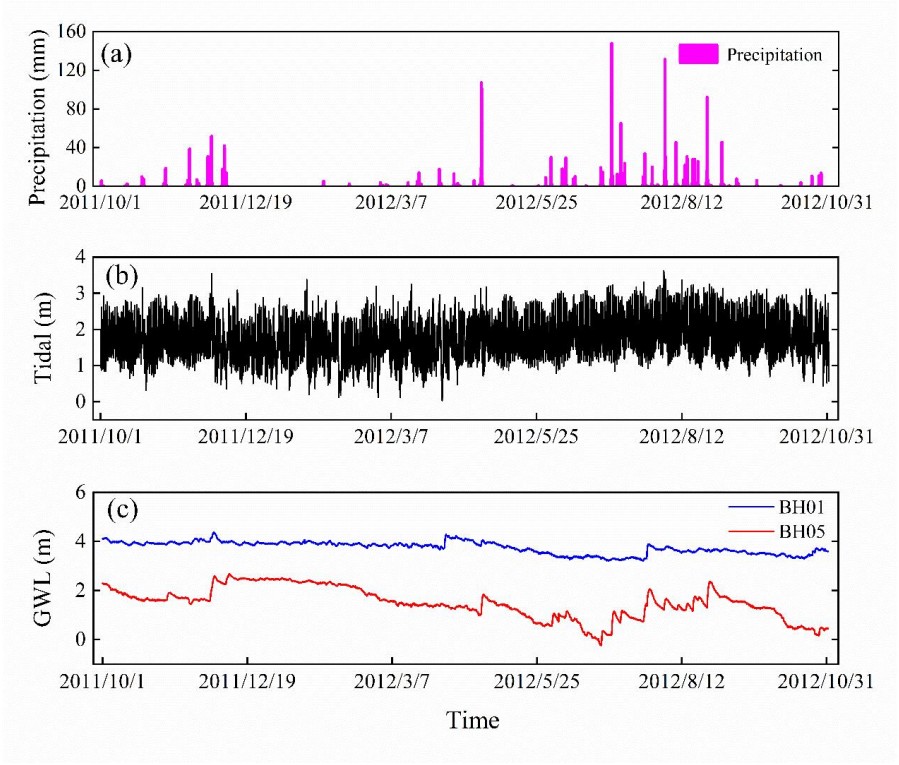






Figure 2. Time-series of the variables in the study, including (a) precipitation, (b) tide,
(c) groundwater level (GWL).

## 3 Methodology

**3.1 Temporal Convolutional Network (TCN)**

The TCN is first proposed by (Lea et al., 2016) for video action segmentation and
detection by hierarchically capturing intermediate feature presentations. Then the term
is extended for sequential data for a wide family of architectures with generic
convolution (Bai et al., 2018; Lea et al., 2017). Suppose that we have an input
hydro-climate sequence at different times $x_0, ..., x_T$, the goal of the modeling is to
predict the corresponding groundwater level as outputs $y_0, ..., y_T$ at each time. The
problem could transfer to build a network $f$ that minimizes the function loss between
observations and actual network outputs $L[(y_0, ..., y_T,),(\hat{y}_0,...,\hat{y}_T)]$, where $\hat{y}_0,...,\hat{y}_T =$
$f(x_0, ..., x_T)$. Currently, a typical TCN consists of dilated, causal 1D full-convolutional
layers with the same input and output lengths. With TCN, the prediction $y_t$ depends
only on the data from $x_0$ and $x_t$ and not include the future data from $x_t$ and $x_T$ (Yan et
al., 2020). With the three key components of TCN, it has two distinguishing
characteristics: 1) the TCN is able to map the same length of output as the input
sequence as in RNN; 2) the convolution involved in TCN is causal, eliminating the
influence of future information on the output.
3.1.1 Causal Dilated Convolutions
In the TCN, the first advantage is accomplished by a 1D full-convolutional
network (FCN) architecture. Different from the traditional CNN, the FCN transforms





the fully connected layers into the convolutional layers for the last layers, preserving
the same length of output as that of the input (Long et al., 2015). As shown in Fig. 3a,
the lengths of the input, the hidden and the output layers are the same in the FCN.
Some zero padding is needed in this step by adding additional zero-valued entries
with a length of kernel size-1 in each layer. The kernel size is the number of
successive elements that are used to produce one element in the next layer.

To avoid the information leakage from the future (after time $t$), the TCN uses

causal convolution instead of standard convolution, where only the elements at or
before time $t$ in the previous layer are adopted into the mapping of the output at time $t$.
Further, the dilated convolution is employed to capture long-term historical
information by skipping a given step size (dilation factor $d$) in each layer. For
example, the dilation factor $d$ increases from 1 to 4 with the evolution of the network
depth ($n$) in an exponential increasing pattern. In this way, a very large receiving
domain is created and all the historical records in the input can be involved in the
prediction model with a deep network.

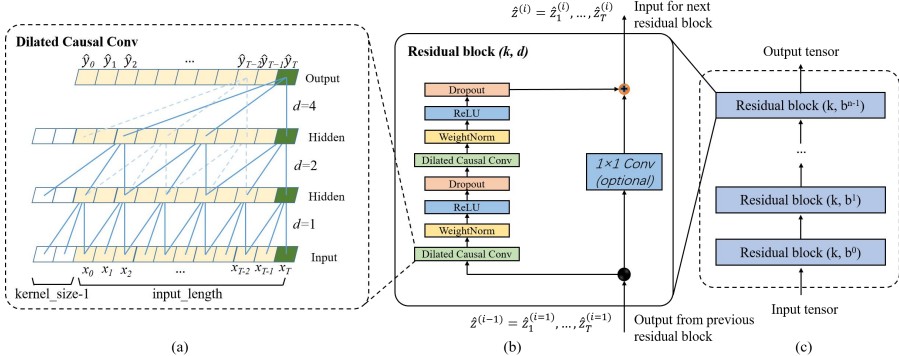


Figure 3. Architectural elements in the proposed TCN. (a) the structure of causal


dilated convolution; (b) the TCN residual block. An 1x1 convolution is added when
residual input and output have different dimensions; (c) framework of residual
connection in the TCN.
3.2.2 Residual Connections
In a high dimensional and long-term sequence, the network structure could be
very deep with increasing complicity and cause a vanishing gradient. To solve this
issue, a residual block structure is introduced to replace the simple 1D causal
convolution layer, so that the designed TCN structure is more generic (He et al.,
2016). The residual block in a TCN is represented in Fig. 3b. It has two convolutional
layers with the same kernel size and dilation factor and non-linearity. To solve
non-linear models, the rectified linear unit (ReLU) is added to the top of the
convolutional layer (Nair and Hinton, 2010). The weight normalization is applied
between the input of hidden layers (Salimans and Kingma, 2016). Meanwhile, a
dropout is added after each dilated convolution for regularization (Srivastava et al.,
2014). For all connected inner residual blocks, the channel widths of input and output
are consistent. But the width may be different between the input of the first
convolutional layer of the first residual block and the output of the second
convolutional layer of the last residual block. Therefore, a 1×1 convolution is added
in the first and last residual block to adjust the dimensions of the residual tensor into
the same. The output of the residual block is represented by $\hat{Z}^{(i)}$ for the $i$th block.
3.2.3 Structure of TCN
A complete structure of TCN is illustrated in Fig.3c. It contains a series of





proceeding residual blocks. The structural characteristics make TCN a deep learning
network model very suitable for complex time-series prediction problems
(Lara-Benítez et al., 2020). The main advantage of TCN is that, similar to RNN, they
have flexible receptive fields and can deal with various length input by sliding
one-dimensional causal convolution kernel. Furthermore, because TCN shares a
convolution kernel and has parallelism, it can process long sequences in parallel
instead of sequential processing like RNN, so it has lower memory usage and shorter
computing time than a cyclic network. Moreover, RNN often has the problems of
gradient disappearance and gradient explosion, which are mainly caused by sharing
parameters in different periods, while TCN uses a standard backpropagation-through-
time algorithm (BPTT) for training, so there is little gradient disappearance and
explosion problem (Pascanu et al., 2012). The detailed mathematical calculation and
associated information of the TCN architecture are referred to (Bai et al., 2018).
**3.2 Long Short-Term Memory network (LSTM)**

LSTM is a special RNN model explicitly designed for long-term dependence

problems. As shown in Fig. 4a, the RNN has a series of repeating modules that
recursively connected in the evolution direction of the sequence. The chain-like
structure permits the RNN to retain important information in a "tanh" layer and
produce the same length of output $\hat{y}_t$ as input $x_t$. However, the short-length
"remember time" is not enough for the groundwater prediction. Especially for our
hourly recorded data, a maximum step about ten reported by (Bengio et al., 1994) is
unable to count the effect of annually, seasonally, and even daily groundwater





variation. Different from the simple layer in the RNN, the LSTM has a more
complicated repeating module with four interacting layers.

The core idea of LSTM is the special structure to control the cell state in the

module as shown in Fig.4b. It includes a cell and an input gate $i_t$, a forget gate $f_t$, and
an output gate $o_t$. The information can directly flow down along cells $C$ without
critical changes, therefore, preserving long-term history messages (Zhang et al.,
2018b). The three gates control which data in a sequence is important to keep or
throw away, and protect the relevant information passed down in the cell to make
predictions. The forget gate $f_t$ has a sigmoid layer to determine which information is
discarded with a value between 0 and 1. The lower the value, the less the information
added to the cell state (Ergen and Kozat, 2018). Opposite the forget gate, the input
gate $i_t$ decides what information to retain in the cell state. It is composed of two parts:
a sigmoid layer and a tanh layer. The two layers are combined to govern which values
will be updated by generating a new candidate value $\tilde{C}_t$. The old cell state $C_{t-1}$ then
can be updated into the new cell state $C_t$ with a weighted function. Finally, the output
gate $o_t$ determines what parts of the cell state should be passed on to the next hidden
state. The detailed calculation of the LTSM can be referenced to (Lea et al., 2016).

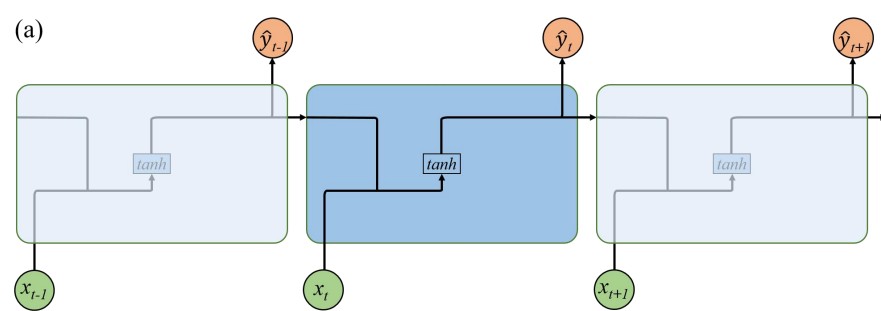






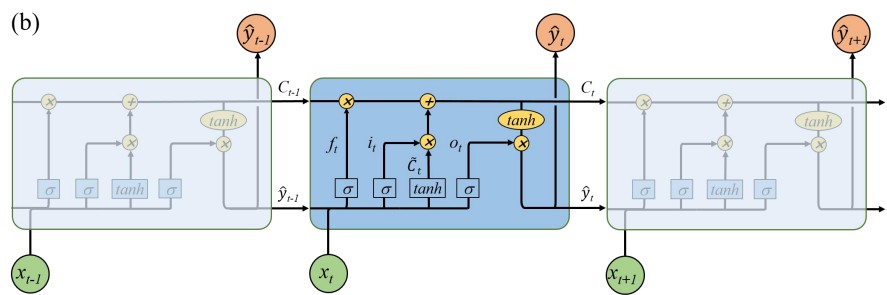


Figure 4. Graphical representation for a) chain like structure of the RNN by assigning

$x_t$ and $\hat{y}_t$ as input and output. The self-connected hidden units allow information to be

passed from one step to the next; b) LSTM's memory block based on RNN. The

hidden block includes three gates (input $i_t$, forget $f_t$, output $o_t$) and a cell state to select

and pass the historical information.

**3.3 Experimental study**

Due to the high complexity of the DL models, setting appropriate

hyper-parameters for the developed networks is very important. Here, the impact of

the size of the input window, the epoch number and the batch size were tested with

different convolutional architectures over the monitoring data (Lara-Benítez et al.,

2020). The learning dataset is first divided into two parts: 80% of the time-series data

is used as training set, and 20% of the data is utilized as testing set. The effect of

different splitting strategies is further tested in section 4. With the increase of the

epoch numbers, the curve gradually approaches to the optimal fitting state from the

initial non-fitting state, but too many epochs frequently lead to over-fitting of the

neural network. Meanwhile, the number of iterations generally increases for updating

weights in the neural network. Therefore, the number of epoch from 0 to 300 is

evaluated. Batch size represents the number of samples between model weight



updates (Kreyenberg et al., 2019). The value of the batch size often is set between 1
and hundreds. Larger batch size often leads to faster convergence of the model, but
may lead to less ideal of the final weight set. To find the best balance between
memory efficiency and capacity, the batch size should be carefully set to optimize the
performance of the network model. Besides these parameters, the number of filters in
the TCN-based and the hidden nodes in the LSTM-based model were as well tested
within reasonable ranges.
The 1-day, 3-, 7-, and 15-days lead prediction experiments were further
conducted to test the capacity of DL methods in predicting long-term groundwater
level in the coastal aquifer. To eliminate the randomness of model training, all
experiments were repeated 5 times and the average values of each index were
compared. In all experiments, the average absolute error (MAE) has been used as the
loss function of networks (Lara-Benítez et al., 2020). The Adam optimizer has an
adaptive learning rate, which can improve the convergence speed of deep networks,
which has been used to train the models (Kingma and Ba, 2015).
**3.4 Evaluation of model performance**
Two evaluation metrics, coefficient of determination ($R^2$) and root mean square
error (RMSE) are selected to quantify the goodness-of-fit between model outputs and
observations ((Zhang et al., 2020)). The two criteria are calculated using the following
equations:
$$RMSE = \sqrt{\frac{1}{N}\sum_{i=1}^{N}(h_i - y_i)^2} \tag{1}$$

$$R^2 = \frac{\sum_{i=1}^{N}(h_i - \bar{h})^2 - \sum_{i=1}^{N}(h_i - y_i)^2}{\sum_{i=1}^{N}(h_i - \bar{h})^2} \tag{2}$$





where $h_i$ is the observed groundwater level at time $i$, $y_i$ is the network prediction
values at time $i$, $\bar{h}$ is mean of the observed groundwater levels, and $n$ is the number
of observations. RMSE measures the prediction precision which creates a positive
value by squaring the errors. The RMSE score is between $[0, \infty]$. If the RMSE
approaches to 0, the model prediction is ideal. $R^2$ measures the degree of model
replication results, ranging between $[-\infty, 1]$. For the optimal model prediction, the
score of $R^2$ is close to 1.
**4 Results and discussions**
**4.1 Hyper-parameter trial experiments**
4.2.1 Experiments of the TCN-based model
The TCN-based model is built on Keras platform, using TensorFlow of python
as the backend. Take the groundwater level dataset in well BH1 as an example, the
trials are set up with a variety combination of different hyper-parameters that are set
in the TCN-based model as illustrated in Table 1. With the fixed number of epoch, the
result of 32 filters is better than that of 16 and 64 filters. Meanwhile, under the
condition of 32 filters, the results of the model decrease with the increase of batch size.
Therefore, when three different batches of 16, 32, and 64 are set for testing, the results
of the 16 batch size of the model are better. Based on the above experimental results,
the influence of different numbers of epoch on the simulation is further explored with
the filters equals to 32 and the batch size equals to 16 as shown in Fig.5. The overall
results of the model are improved when the number of epoch increases from 100 to
190 though the variation is not strictly linear, and the results turn stable with minor




fluctuations when the number of epoch exceeds 200.
Table 1. The RMSE and $R^2$ values between the observed and predicated groundwater
levels in well BH1 with different numbers of epochs, different numbers of filters, and
different batch sizes. The bold values represent the optimal hyper-parameters with the
smallest RMSE and the highest $R^2$ scores in the TCN-based model.

| Epoch | filters | Batch size | RMSE(m) | $R^2$ | Time(min) |
|---|---|---|---|---|---|
| 100 | 32 | 16 | 0.0182 | 0.9904 | 1.29 |
| | | 32 | 0.0117 | 0.9876 | 1.05 |
| | | 64 | 0.0117 | 0.9875 | 0.78 |
| 200 | 16 | 16 | 0.0078 | 0.9946 | 2.41 |
| | | 32 | 0.0068 | 0.9959 | 1.75 |
| | | 64 | 0.009 | 0.9942 | 1.19 |
| **200** | **32** | **16** | **0.0059** | **0.997** | 2.58 |
| | | 32 | 0.0075 | 0.9948 | 2.01 |
| | | 64 | 0.0082 | 0.9938 | 1.51 |
| 200 | 64 | 16 | 0.0125 | 0.9906 | 3.68 |
| | | 32 | 0.0101 | 0.9907 | 3.21 |
| | | 64 | 0.0157 | 0.9775 | 2.76 |
| 300 | 32 | 16 | 0.0065 | 0.9955 | 3.8 |
| | | 32 | 0.0076 | 0.9946 | 3.01 |
| | | 64 | 0.0099 | 0.9904 | 2.22 |


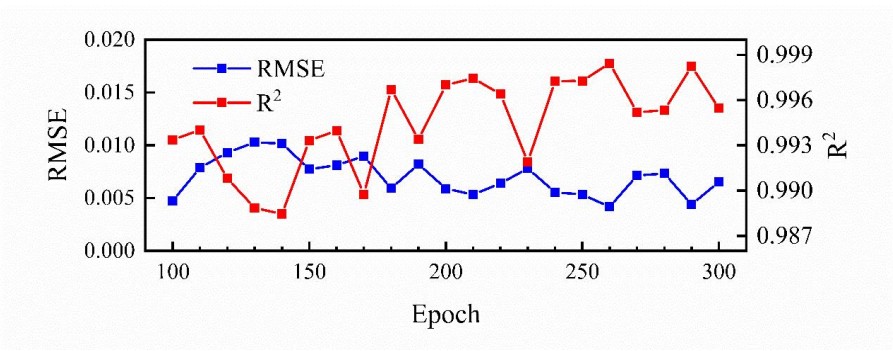




Figure 5. The variation of RMSE and $R^2$ values between the observed and simulated
groundwater levels of well BH1 with the increasing number of epoch when the
number of filters is 32 and the batch size is 16.
**4.2.2 Experiments of the LSTM-based model**
The maximum epoch and the number of hidden nodes are two key parameters
affecting the simulation accuracy of LSTMs (Zhang et al., 2018a). Different
hyper-parameter combinations are tested as well as in the proposed TCN-based model
with groundwater levels in well BH1. The RMSE, $R^2$ and running time are shown in
Table 2. With fixed number of hidden nodes, the results of 100 and 200 epochs are
better than that in the 300 epochs experiment. A detailed variation of RMSE and $R^2$
values with increasing hidden nodes and epoch are further illustrated in Fig. 6. The
figure shows that the RMSE and $R^2$ have a decreasing and increasing trend separately
when number of epochs is greater than 150 but they turn to the opposite way when it
is larger than 240. The variations of RMSE and $R^2$ with increasing hidden nodes have
similar changes as well. The results indicate that though an insufficient number of
neurons may decrease the learning ability of the network, an increasing training
hyper-parameters may not ensugare better rFesults.
Table 2. The RMSE and $R^2$ values between the observed and simulated groundwater
levels in well BH1 with different numbers of epochs and hidden nodes. The bold
values represent the optimal hyper-parameters used in the proposed LSTM-based
model.

| Epoch | Hidden nodes | RMSE | $R^2$ | Time(min) |
|-------|--------------|------|-------|-----------|
| 100 | 50 | 0.0104 | 0.9902 | 1.01 |



| | | | |
|---|---|---|---|
| | 60 | 0.0098 | 0.9916 | 1.38 |
| | 70 | 0.0095 | 0.9922 | 1.53 |
| | 80 | 0.01 | 0.9913 | 1.75 |
| | 50 | 0.0094 | 0.9922 | 1.91 |
| **200** | 60 | 0.0089 | 0.9931 | 2.59 |
| | **70** | **0.0088** | **0.9932** | **2.96** |
| | 80 | 0.0092 | 0.9925 | 3.28 |
| | 50 | 0.0101 | 0.9903 | 2.86 |
| 300 | 60 | 0.0105 | 0.9901 | 3.85 |
| | 70 | 0.0103 | 0.9907 | 4.29 |
| | 80 | 0.0120 | 0.9872 | 4.92 |


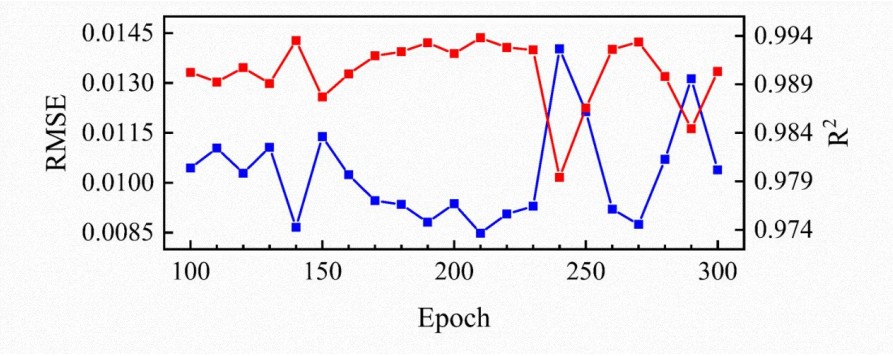


Figure 6. The variation of RMSE and $R^2$ values between the observed and simulated
groundwater levels of well BH1 with the increasing of the number of epochs when the
hidden node is 50.
The trial experimental results present similar fitting pattern shared by the two
kind of networks. Inadequate hyper-parameters often leads to deficient learning
ability of the network. In the contrary, excessive parameter setting may cause neural
network overfitting issues. In addition, the growing parameters dramatically increase
the computational cost in the network. For example, the time cost from 50 to 80
hidden nodes increased about 1.7 times in each iteration trial in the LSTM-based





model. Therefore, during implementation, 200 epochs, 32 filters, and the 16 batch size
were chosen as the optimal parameters in the TCN network. For the LSTM network,
the number of epoch and hidden nodes were chosen as 200 and 70.
**4.3 Model performance and evaluation**
The optimal hyper-parameters of the proposed TCN-based model for groundwater
level predicting are shown in Table 1 (epoch = 200, filters = 32 and batch size = 16).
Besides that, the kernel size in each convolutional layer is set as 6, the dilations are
[1,2,4,8]. For the LSTM-based model, the batch size is set to 148 with epoch=200 and
nodes=70. The same hyper-parameters are then utilized to construct TCN and LSTM
architectures for prediction of groundwater level in different monitoring wells.
The simulated groundwater level in the training and testing stages by the two
models are shown in Fig. 7. For both models, the simulated values completely capture
the variation of groundwater levels in monitoring wells with overlapped plot. The $R^2$
and RMSE values of simulation results are listed in Table 3. For the TCN-based
model, the values of RMSE are 0.0019 and 0.0166 for BH1, and the values of $R^2$ are
larger than 0.999 in the prediction. For the LSTM-based model, the RMSE values are
0.0074 and 0.0588, and the $R^2$ values are 0.9957 and 0.9980. These metrics indicate
that both of the models can "remember" the historical records and produce true
observations. The simulation accuracy of TCN-based models is slightly higher than
the LSTM-based models. In addition, the running time of the TCN-based model is 2.6
minutes, which is faster than that of the TCN-based model by eliminating the gate
selection.



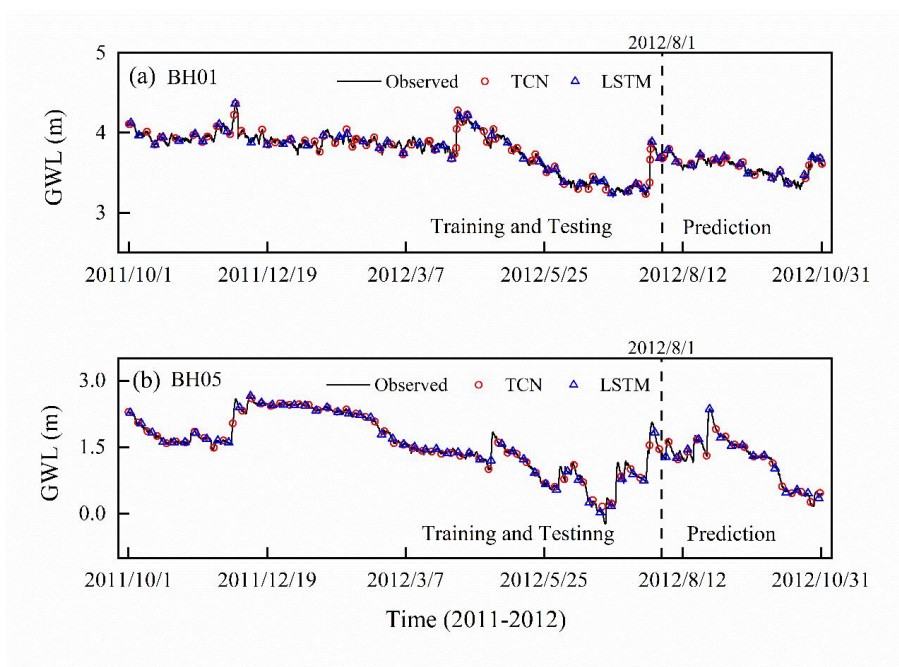


Figure 7. The simulation results of groundwater level of different monitoring wells by

TCN-based model. The black dash line divides the data into two groups: the training

and testing set.


Table 3. The model results for groundwater level in the training and testing and

prediction stage

| Well | Model | Training and Testing | | | Prediction | | |
|---|---|---|---|---|---|---|---|
| | | MAE | RMSE | $R^2$ | MAE | RMSE | $R^2$ |
| BH01 | TCN | 0.0017 | 0.0068 | 0.9992 | 0.0009 | 0.0019 | 0.9997 |
| | LSTM | 0.0053 | 0.0077 | 0.9990 | 0.0050 | 0.0074 | 0.9957 |
| BH05 | TCN | 0.0070 | 0.0279 | 0.9981 | 0.0061 | 0.0166 | 0.9990 |
| | LSTM | 0.0082 | 0.0116 | 0.9997 | 0.0168 | 0.0558 | 0.9980 |


**4.4 Long term leading time prediction**

The TCN- and LSTM-based models were further adjusted to predict the groundwater level of the coastal aquifer over three months ahead with different leading period. Prediction results of groundwater level with 1-day, 3-, 7-, and 15-days leading time of TCN- and LSTM-based models are illustrated in Fig. 8 and Fig. 9 for wells BH1 and BH5 respectively. The results show that the predicted groundwater values in monitoring wells have the same change trend as the actual groundwater level. Both of the models are able to capture the variation trend of groundwater levels in the two monitoring wells.

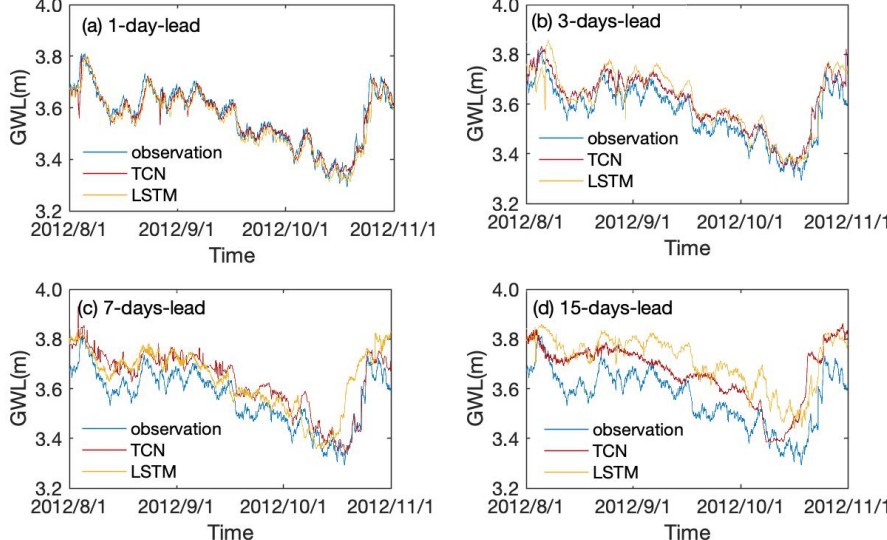

Figure 8. The observed and prediction values of the groundwater level with TCN- and LSTM-based models for 1-day, 3-, 7- and 15-days lead period in monitoring well BH01.



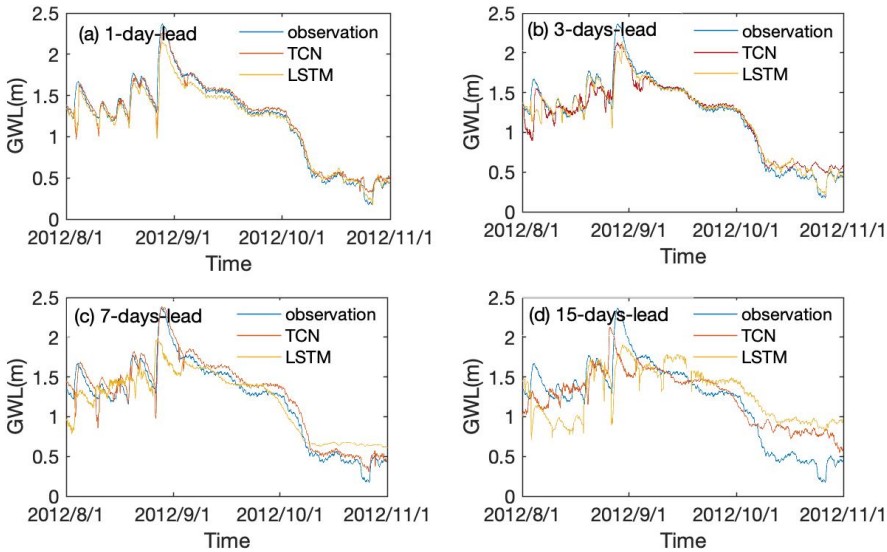

Figure 9. The observed and prediction values of the groundwater level with TCN- and LSTM-based models for 1-day, 3-, 7- and 15-days lead period in monitoring well BH05.

To quantitatively compare the prediction accuracy of the proposed TCN- and LSTM-based models, the results of two evaluation metrics with the model running time in different monitoring wells are summarized in Table 4. It can be learned that the $R^2$ value of TCN-based models decreased from 0.9386 to 0.1406 for well BH01 and from 0.9670 to 0.7271 for well BH05. Correspondingly, an increase of RMSE values from 0.028 to 0.1209 and 0.0934 to 0.206 are observed for BH01 and BH05, separately. A similar variation pattern is recognized for LSTM-based model with smaller $R^2$ and higher RMSE than that of the TCN-based model. While, the average running time of TCN-based is about 3.4 seconds, which is about 6 seconds for LSTM-based models.





Table 4. The model results for groundwater level in the long term prediction

| Well | Model | Prediction | | Model | Prediction | |
|------|-------|------|------|-------|------|------|
| | | RMSE | $R^2$ | | RMSE | $R^2$ |
| BH01 | TCN-1 | 0.0280 | 0.9386 | LSTM-1 | 0.0349 | 0.9047 |
| | TCN-3 | 0.0550 | 0.7638 | LSTM-3 | 0.0640 | 0.6802 |
| | TCN-7 | 0.0741 | 0.5713 | LSTM-7 | 0.0956 | 0.2874 |
| | TCN-15 | 0.1209 | -0.1407 | LSTM-15 | 0.1486 | -0.7227 |
| BH05 | TCN-1 | 0.0934 | 0.9670 | LSTM-1 | 0.1012 | 0.9613 |
| | TCN-3 | 0.1375 | 0.9285 | LSTM-3 | 0.1086 | 0.9554 |
| | TCN-7 | 0.1084 | 0.9296 | LSTM-7 | 0.2050 | 0.8406 |
| | TCN-15 | 0.2060 | 0.7271 | LSTM-15 | 0.3515 | 0.5330 |


The results showed that the TCN- and LSTM-based models are able to predict the

variation of groundwater levels with longer leading period more than one time step.
The performance of the two networks were further evaluated with Taylor diagrams by
taking different criteria aspects into account (Taylor, 2001). The comparisons of
TCN- and LSTM-based model are shown in Fig. 10. As the metrics distributed away
from the reference point (Ref), the deviation of prediction from observation is
gradually increased with extending of leading period. Taken well BH01 for example,
the prediction with 1-day (24 hours prediction window) in advance are the highest in
agreement with the actual situation in the two models. The two simulation results
have the lowest RMSE values and highest $R^2$ values for both models. The prediction
precision gradually decreases with the extending of leading time. For the leading time
smaller than 7-days, 168 time steps prediction in advance, the evaluation metrics have
acceptable values of less than 0.1 for RMSE but the $R^2$ values have been greatly
dropped. For the 15-days (360 time steps) leading period, the RMSE of the TCN- and
LSTM-based models have increased to 0.1209 and 0.1486 with negative $R^2$ values,
which suggest a kind of overestimation in well BH01.

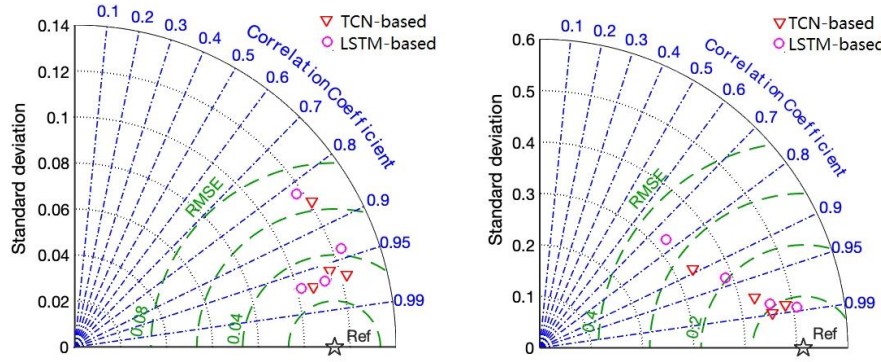


Figure 10. Taylor diagrams with statistical (RMSE, correlation coefficient, and
standard deviation) comparison results of the TCN-based and LSTM-based models
for well (a) BH01 (b) BH05.

Overall, TCN- and LSTM-based models both have strong prediction ability. The

performance of the TCN-based model is better than that of the LSTM-based model in
the three months prediction concerning both model precision and running time.
However, the model precision decreases when the leading period is increasing. The
causal dilated convolutions used by TCNs are better at capturing long-term
dependencies of time series data than recurrent units, improving the efficiency of
neural networks and shortening the network running time. The TCN-based models are
able to provide accurate predictions once they are trained. As expected, the processing
speed of parallel convolution TCN-based models for long input sequences is faster
than that of recurrent networks. This seems to be a basic advantage of real-time
monitoring and early warning. In real-time monitoring and early warning, it is
necessary to obtain predictions quickly to make wise decisions.





### 4.5 Influence of training set percentage

In the following section, we discuss the similarities and differences between TCN- and LSTM-based in terms of training set percentage. As we all know, data-driven methods are supported by data; however, how much data is needed to build an effective model is still a problem. This is because specific problems depend on application cases, data features, and model features (Wunsch et al., 2021). In our study, the data is the hourly-monitored data from 2011 to 2012. From 2011, we set 20%, 30% to 90% training sets in turn, so as to gradually expand the length of training set.

Fig. 11 shows the effect of increasing the percentage of training set on the performance of the model. All experiments were repeated five times, and the average values of each index were compared to make them comparable. We observed that the overall performance of the TCN-based model improved with the increase of the percentage of training set. When the training set reached 80%, the performance was relatively optimal, and then the performance began to deteriorate with the increase of the percentage of training set; at the same time, it can be seen that the performance of the LSTM-based model tends to be stable when the training set reaches 70%, and then decreases slightly with the increase of training set. Therefore, it is not that the more training sets, the better the performance of the model. We should carefully evaluate and shorten the training data set as much as possible when necessary. Finally, we set 80% of the training set length to simulate the coastal aquifer time-series data.

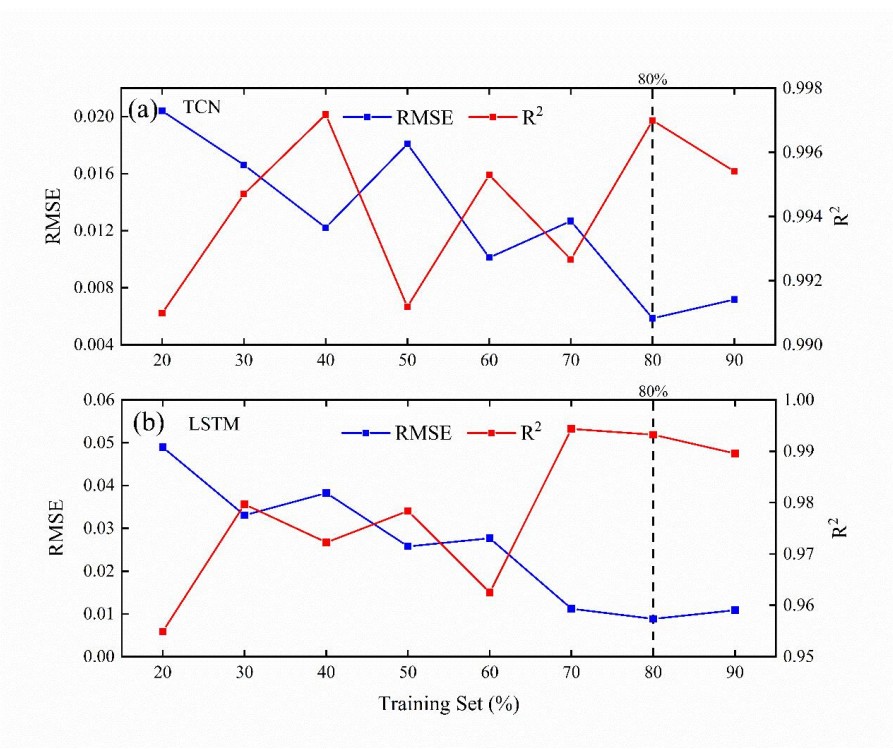


Figure 11. Influence of the percentage of training set on the performance of the model
**5 Conclusions**

A TCN-based deep learning model is proposed in this paper to predict

groundwater levels in coastal aquifers. Hyper-parameter searches was first conducted
and several different TCN-based models were tested to obtain a good architecture
configuration. The results indicate that a deeper, broader model does not necessarily
guarantee better predictions. The optimal configuration then were adopted for the
networks of all monitoring data. This means that different data could share the same
network architecture without adjusting in each case and broaden its application in
different areas. With comparison to observations, the TCN-based model has achieved
satisfactory performance on the prediction of groundwater levels, which can well



capture the fluctuation of water level and provide possible saltwater intrusion
information in the coastal area. Thus, it can be used as a new promising method for
time-series prediction of hydrogeological data especially when the regional data is
difficult to collect in a complex system.
To validate the newly developed TCN-based model, its performance is compared
with the LSTM-based recurrent networks. The TCN-based model outperforms the
LSTM-based model in view of both accuracy and efficiency. Meanwhile, three
months ahead predictions were conducted with different leading periods. A
decreasing precision is revealed when the leading time increases. In particular, once
TCN was trained, due to the use of parallel convolution to process the input sequence,
its prediction speed is significantly faster than recurrent networks. In summary, our
research shows that TCN is a very powerful alternative to the LSTM network. It can
provide accurate predictions and is suitable for more complex real-time applications
because of its high efficiency.
**Acknowledgements**
This work was jointly supported by National Natural Science Foundation of
China (No: 41702244), the Program for Jilin University (JLU) Science and
Technology Innovative Research Team (No. 2019TD-35).
**Code availability**
The pieces of code that were used for all analyses are available from the authors
upon request.
**Data availability**



The data sets that have been analyzed in this paper are available from the
authors upon request.
**Author contribution**
XZ drafted the manuscript and revised the manuscript. GC designed the
experiments and collected all the data. DF developed the model code and performed
the simulations. ZD was responsible for the project design, oversaw the analysis, and
conducted manuscript revision as the project leader and the senior scientist.
**Competing interests**
The authors declare that they have no conflict of interest.

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

Proof‐of‐Concept Study of Using a Less Permeable Slice Along the
Shoreline to Increase Fresh Groundwater Storage of Oceanic Islands:
Analytical and Experimental Validation, Water Resour. Res., 55(8),
6450-6463, https://doi.org/10.1029/2018wr024529, 2019.
Maier, H. R., and Dandy, G. C.: Neural networks for the prediction and forecasting of
water resources variables: a review of modelling issues and applications,
Environ. Modell. Softw., 15(1), 101-124,
https://doi.org/10.1016/S1364-8152(99)00007-9, 2000.
Mei, Y., Tan, G., and Liu, Z.: An improved brain-inspired emotional learning





algorithm    for    fast    classification,    Algorithms,    10(2),    70,
https://doi.org/10.3390/a10020070, 2017.

Nair, V., and Hinton, G. E.: Rectified linear units improve Restricted Boltzmann
machines,   In   Proceedings   of   the   27th   International   Conference   on
International    Conference    on    Machine    Learning,    pp.    807-814,
https://dl.acm.org/doi/10.5555/3104322.3104425, 2010.

Park, Y., Lee, J. Y., Kim, J. H., and Song, S. H.: National scale evaluation of
groundwater   chemistry   in   Korea   coastal   aquifers:   evidences   of   seawater
intrusion,    Environ.    Earth    Sci.,    66(3),    707-718,
https://doi.org/10.1007/s12665-011-1278-3, 2011.

Pascanu, R., Mikolov, T., and Bengio, Y.: On the difficulty of training recurrent
neural    networks,    Machine    Learning,    pp.    2347-2355,
https://doi.org/10.48550/arXiv.1211.5063, 2012.

Pratheepa, V., Ramesh, S., Sukumaran, N., and Murugesan, A. G.: Identification of
the sources for groundwater salinization in the coastal aquifers of Southern
Tamil    Nadu,    India,    Environ.    Earth    Sci.,    74(4),    2819-2829,
https://doi.org/10.1007/s12665-015-4303-0, 2015.

Rumelhart, D. E., Hinton, G. E., and Williams, R. J.: Learning representations by
back-propagating    errors,    Nature,    323(6088),    533-536,
https://doi.org/10.1038/323533a0, 1986.

Sahoo, S., Russo, T. A., Elliott, J., Foster, I., and Argonne National Lab, A. I. L.:
Machine   learning   algorithms   for   modeling   groundwater   level   changes   in
agricultural regions of the U.S, Water Resour. Res., 53(5), 3878-3895,
https://doi.org/10.1002/2016WR019933, 2017.

Salimans, T., and Kingma, D. P.: Weight normalization: A simple reparameterization
to   accelerate   training   of   deep   neural   networks,   Advances   in   Neural
Information    Processing    Systems,    pp.    901-909,
https://doi.org/10.48550/arXiv.1602.07868, 2016.

Senthil Kumar, A. R., Sudheer, K. P., Jain, S. K., and Agarwal, P. K.: Rainfall-runoff
modelling   using   artificial   neural   networks:   comparison   of   network   types,





706   Hydrol. Process., 19(6), 1277-1291, https://doi.org/10.1002/hyp.5581, 2005.

707 Seo, Y., Kim, S., Kisi, O., and Singh, V. P.: Daily water level forecasting using

708   wavelet decomposition and artificial intelligence techniques, J. Hydrol., 520,

709   224-243, https://doi.org/10.1016/j.jhydrol.2014.11.050, 2015.

710 Solgi, R., Loáiciga, H. A., and Kram, M.: Long short-term memory neural network

711   (LSTM-NN) for aquifer level time series forecasting using in-situ piezometric

712   observations, J. Hydrol., 601, https://doi.org/10.1016/j.jhydrol.2021.126800,

713   2021.

714 Srivastava, N., Hinton, G., Krizhevsky, A., Sutskever, I., and Salakhutdinov, R.:

715   Dropout: A simple way to prevent neural networks from overfitting, J. Mach.

716   Learn. Res., 15(1), 1929-1958,

717   https://dl.acm.org/doi/abs/10.5555/2627435.2670313, 2014.

718 Taylor, K. E.: Summarizing multiple aspects of model performance in a single

719   diagram, J. Geophys. Res.-Atmos., 106, 7183-7192,

720   https://doi.org/10.1029/2000JD900719, 2001.

721 Torres, J. F., Troncoso, A., Koprinska, I., Wang, Z., and Martínez-Álvarez, F.: Deep

722   Learning for Big Data Time Series Forecasting Applied to Solar Power, The

723   13th International Conference on Soft Computing Models in Industrial and

724   Environmental Applications, pp. 123-133,

725   https://doi.org/10.1007/978-3-319-94120-2_12, 2018.

726 Wan, R., Mei, S., Wang, J., Liu, M., and Yang, F.: Multivariate temporal

727   convolutional network: A deep neural networks approach for multivariate time

728   series forecasting, Electronics, 8(8),

729   https://doi.org/10.3390/electronics8080876, 2019.

730 Wunsch, A., Liesch, T., and Broda, S.: Groundwater level forecasting with artificial

731   neural networks: a comparison of long short-term memory (LSTM),

732   convolutional neural networks (CNNs), and non-linear autoregressive

733   networks with exogenous input (NARX), Hydrol. Earth Syst. Sci., 25(3),

734   1671-1687, https://doi.org/10.5194/hess-25-1671-2021, 2021.

735 Xu, Z., and Hu, B.X.: Development of a discrete-continuum VDFST-CFP numerical





model for simulating seawater intrusion to a coastal karst aquifer with a
conduit system, Water Resour. Res., 53(1), 688-711,
https://doi.org/10.1002/2016wr018758, 2017.

Xue, Y., Wu, J., Ye, S., and Zhang, Y.: Hydrogeological and hydrogeochemical
studies for salt water intrusion on the south coast of Laizhou Bay, China,
Groundwater, 38(1), 38-45,
https://doi.org/10.1111/j.1745-6584.2000.tb00200.x, 2000.

Yan, J., Mu, L., Wang, L., Ranjan, R., and Zomaya, A. Y.: Temporal Convolutional
Networks for the Advance Prediction of ENSO, Sci Rep., 10(1), 8055-8055,
https://doi.org/10.1038/s41598-020-65070-5, 2020.

Zeng, X., Wu, J., Wang, D., and Zhu, X.: Assessing the pollution risk of a
groundwater source field at western Laizhou Bay under seawater intrusion,
Environ. Res., 148, 586-594, https://doi.org/10.1016/j.envres.2015.11.022,
2016.

Zhan, C., Dai, Z., Soltanian, M. R., and Zhang, X.: Stage‐wise stochastic deep
learning inversion framework for subsurface sedimentary structure
identification, Geophys. Res. Lett., 49(1), e2021GL095823,
https://doi.org/10.1029/2021GL095823, 2022.

Zhang, D., Lin, J. Q., Peng, Q. D., Wang, D. S., Yang, T. T., Sorooshian, S., Liu, X.
F., and Zhuang, J. B.: Modeling and simulating of reservoir operation using
the artificial neural network, support vector regression, deep learning
algorithm, J. Hydrol., 565, 720-736,
https://doi.org/10.1016/j.jhydrol.2018.08.050, 2018a.

Zhang, J., Zhang, X. Y., Niu, J., Hu, B. X., Soltanian, M. R., Qiu, H., and Yang, L.:
Prediction of groundwater level in seashore reclaimed land using wavelet and
artificial neural network-based hybrid model, J. Hydrol., 577, 123948,
https://doi.org/10.1016/j.jhydrol.2019.123948, 2019.

Zhang, J., Zhu, Y., Zhang, X., Ye, M., and Yang, J.: Developing a Long Short-Term
Memory (LSTM) based model for predicting water table depth in agricultural
areas, J. Hydrol., 561, 918-929, https://doi.org/10.1016/j.jhydrol.2018.04.065,



2018b.

Zhang, X. Y., Dong, F., Dai, H., Hu, B. X., Qin, G. X., Li, D., Lv, X. S., Dai, Z. X.,

and Soltanian, M. R.: Influence of lunar semidiurnal tides on groundwater

dynamics in estuarine aquifers, Hydrogeol. J., 28(4), 1419-1429,

https://doi.org/10.1007/s10040-020-02136-8, 2020.

Zhang, X. Y., Miao, J. J., Hu, B. X., Liu, H. W., Zhang, H, X., and Ma, Z.:

Hydrogeochemical characterization and groundwater quality assessment in

intruded coastal brine aquifers (Laizhou Bay, China), Environ. Sci. Pollut.

Res., 24(26), 21073-21090, https://doi.org/10.1007/s11356-017-9641-x, 2017.
