# Peer review of "Advance prediction of coastal groundwater levels with temporal"

_Hydrology and Earth System Sciences, 2022_

## Author Response (AR1)

**Responses to review comments for Manuscript #Hess-2022-236 "Advance prediction of coastal groundwater levels with temporal convolutional network."**

Dear Editors and Reviewers:

Thanks for your constructive comments. Those comments are all valuable and very helpful for revising and improving our manuscript. We have studied these comments carefully and revised the manuscript accordingly. The responses to reviewers' comments are listed below sequentially. The revisions in the tracking manuscript are highlighted in red.

**Reviewer 1:**

*Major points:*

*1. Based on the platform information, the efficiency of the two developed DL methods needs to be checked. They often take minutes to hours to complete a network construction, especially for the advance prediction part.*

**Response**: Thanks for the suggestions. We have checked the training and predicting time for both models and modified the corresponding parts in Section 4 and Table 4.

*2. The manuscript built two models and compared their results simultaneously, it would be better to include the LSTM in the title and further clarify the performance of the two models in the abstract and discussion part.*

**Response**: Thanks for the suggestion. We agree with the reviewer's comment. The two models are actually both good at predicting variables in long-term time series. The LSTM was the most widely used approach and the TCN was a new framework and has not been used in the groundwater region. The results show comparable results between the two models. Based on the suggestion, we have modified our title into "Advance prediction of coastal groundwater levels with temporal convolutional and long short-term memory network" by including LSTM network. We as well adjusted the introduction, highlights and discussion to better state the results of LSTM. A statistical t-test was as well conducted to determine the significance of the difference.

*3. Minor points: Highlights: please check if the monitoring wells are all located in the same aquifer and be consistent through the paper.*

**Response**: Thanks for the comment. We have checked the depth of the monitoring wells in the geological cross section, the monitoring wells are all located in the same aquifer. We have modified all the corresponding parts that related to "aquifer" through the paper (line 13, line 501).

*4. Line 27: a full name of TCN is needed here.*

**Response**: Suggestion followed. A full name of TCN, temporal convolutional network has been added in the abstract.

*5. Line 29: change the "first" to "beginning", and the following three months data.*

**Response**: Suggestion followed.

*6. Line 31: Please check the time steps "24, 72, 18 and 360 time steps in advance.". It should have an increasing order.*

**Response**: Thanks for the comment. It should be 180 time steps. We have modified it and added the corresponding days to the time steps.

*7. Line 32: The sentence is redundant with two "prediction" statements. And why only the one time step result is stated here, please modify the words. All the results from different leading periods should come together. Meanwhile, please correspond the time steps to the real time when talking about leading periods.*

**Response**: Thanks for the comment. As we have 2 wells with 5 different leading periods including the one step prediction, so we have removed the detailed information for one time step parts and stated the results with the advanced prediction together.

*8. Line 54: China experiences critical saltwater intrusion as well and has great research on this area, please check the paper (you could add two papers that you want*

*us to cite).*

**Response**: Thanks for the comment, we agree with the reviewer. Two papers that related to the serious saltwater intrusion have been added here.

*9. Line 62-64: This sentence does not make sense, please check.*

**Response**: Suggestion followed. The sentence has been revised.

*10. Line 117: delete "several"*

**Response**: Suggestion followed.

*11. Line 128: change "prediction" to "predict"*

**Response**: Suggestion followed.

*12. Line 132: change "concept" to "concepts"*

**Response**: Suggestion followed.

*13. Line 140: delete "basically"*

**Response**: Suggestion followed.

*14. Line 144: change "have been" to 'were"*

**Response**: Suggestion followed.

*15. Line 150: please make sure that you are describing the data is integrate and how the wells are distributed in this area.*

**Response:** Suggestion followed. The sentence has been revised to state an integrate data.

*16. Line 159: change "three wells" to "two wells"*

**Response:** Thanks for pointing this out. Suggestion followed.

*17.Line 160: Please check the total real available data items in this area, as the precipitation is daily monitored.*

**Response:** Thanks for the comment. We have recalculated the data size. The groundwater level and tidal were hourly monitored within 13 months, and the precipitation, as the reviewer indicated, is daily monitored. Therefore, the total collected data was 9480*3+396=28,836.

*18. Line 366: please check the typo error "may not ensugare better rFesults."*

**Response:** Suggestion followed.

*19. Line 392: The Fig.7 includes prediction stages as well. Please is the description "The simulated groundwater level in the training and testing stages" correct.*

**Response:** Thanks for pointing this out. We have added the prediction stage in this sentence. Meanwhile, we further clarified the results were for the prediction stage in the same paragraph.

*20. Line 396: "the values of RMSE are 0.0019 and 0.0166 for BH1", I only found one well for two values, which needs to be checked.*

**Response:** Thanks for pointing this out. The results were for BH01 and BH05 separately.

*21. Line 460: It would be better if you add the leading periods with the markers in Fig.10. Then it can clearly show the precision of each model.*

**Response:** Thanks for the suggestion. We have added the leading time in the new

revised Figure 10.

**Reviewer 2:**

*This manuscript presents an innovative and practical approach to predict the coastal groundwater levels. The authors developed a TCN-based model to predict coastal groundwater levels and compared the results to the existing popular LSTM model. These methods have promising application in real-time prediction of hydrogeological data. As the authors argue, it is essential to achieve single step real-time prediction. The novelty part is the discussion of advanced prediction, which is necessary as sometimes we are more concern about the variation in a longer time. The work presented in this manuscript can have a substantial contribution to the studies of coastal groundwater levels' prediction. Overall, the contents of the manuscript are interesting. Logicality of the paper is clear, and the results are well discussed and explained. However, there are some issues should be better explained after I had read through the paper.*

**Response:** Thanks very much for the overall positive comment on this manuscript.

*Major comments:*

*(1) Are BH01 and BH05 predictions using the same TCN and LSTM models? Or the prediction models for different wells need to be trained separately, and each well needs to use the model trained on its own data for prediction. It is recommended that the authors clarify this point to help the reader better understand the overall forecast implementation process.*

**Response:** Thanks for the comment. Yes, the TCN and LSTM were trained separately for the two wells. The framework of the models were the same, but the hyper-parameters were different such as epochs, filters and batch size. We have clarified this in the Method section in line 285.

*(2) During the hyperparameter comparison process, different models are evaluated based on the training set or the test set. If the evaluation is based on the training set, whether the performance of the test set is consistent with the training set under different hyperparameters.*

**Response:** Thanks for the comment. The hyper-parameter set was evaluated with the prediction dataset. We have added this in the line 331. Meanwhile, the corresponding label of figures and tables have been adjusted as well. Further, the performance of the test set is consistent with the training set.

*(3) The conclusion should be careful on the comparison results of LSTM and TCN.*

**Response:** Thanks for the suggestion. We have carefully compared the results from TCN and LSTM. In the revised manuscript, the efficiency and accuracy are separately compared. Meanwhile, we have added the running time of the advance prediction in Table 4. Further, we have revised the corresponding parts in lines 439-442 and in lines 436-468.

*Minor comments:*

*(1) Abstract: The full name of TCN should be defined before using this abbreviation.*

**Response:** Thanks for the suggestion. The full name has been added in line 27 in the abstract.

*(2) Line 42-43: "… localized groundwater prediction…" should be "… localized groundwater level prediction…"?*

**Response:** Suggestion followed. It has been added in line 42.

*(3) Line 94, Here the BP neural network is first mentioned. An explanation is needed here or add the relationship between BP and other networks that mentioned before.*

**Response:** Thanks for the suggestion. Different from the RNN that has a recursion in evolution direction, the BP is flows one way and uses error backpropagated from the last to the first time step to adjust the weights of neural. The property of BP has been added in line 95.

*(4) Line 123, a reference is need before "Therefore"*

**Response:** Suggestion followed in line 124.

*(5) Line 123, have the longer periods prediction have been used in other area?*

**Response:** Thanks for the suggestion. Researches on the application of TCN have been mainly adopted in climate science and we have added the information line line 122-124.

*(6) Line 317, typo ")"*

**Response:** Thanks for pointing this out. Suggestion has been followed.

*(7) The detailed form of the data that the neural network used is needed to further illustrate the structure of the network.*

**Response:** Thanks for the suggestion. In the manuscript, $y$ represents the output, groundwater level, and $x$ represents the input, including tidal, precipitation and groundwater levels. The information has been clarified in line 161 and 188.

*(8) In the methodology, the tidal, precipitation and groundwater level are corresponding which variables in the equation.*

**Response:** Thanks for the suggestion. We have adjusted the variables in Eq.1, the $y$ is changed to $x'$ to eliminate the confusion between the output and normalized input.

*(9) Figure 6: The author needs to add the legends corresponding to the different colors to avoid confusion to the reader*

**Response:** Thanks for the suggestion. The legend have added in Fig. 10 to show the different markers from TCN- and LSTM-based model.

---

## Author Response (AR2)

**Responses #2 to review comments for Manuscript #Hess-2022-236 "Advance prediction of coastal groundwater levels with temporal convolutional network."**

Dear Editors and Reviewers:

Thanks for your constructive comments. We have addressed these comments carefully and revised the manuscript accordingly. The responses to reviewers' comments are listed below sequentially. The revisions in the tracking manuscript are highlighted in red. The grammar has been as well checked and revised through the manuscript.

**Reviewer 1:**

*Accepted as is.*

**Response**: Thanks very much for the review. We are glad that the paper could be accepted.

**Reviewer 2:**

*Comments to "Advance prediction of coastal groundwater levels with temporal convolutional and long short-term memory networks"*

*This study evaluated two DL algorithms (TCN and LSTM) in predicting the groundwater levels of a coastal aquifer in China, Laizhou Bay. The results demonstrated that both models showed great ability to learn complex patterns in advance using historical data with different leading periods. By comparing the simulation accuracy and efficiency, the TCN-based model slightly outperformed the LSTM-based model but less efficient in training time.*

**Response**: Thanks very much for the overall positive review of this paper.

*I think this paper is well organized and easy to read, and the conclusion is well supported by the results. This is a good work in groundwater prediction. In addition, the authors have made sufficient revisions in accordance to reviewers' comments. However, I still have a small suggestion to increase the readability, the authors could*

*have a discussion on the influence of human activities which are not considered in this study (e.g., groundwater pumping for agriculture activities) on the performances of two DL methods.*

**Response**: Thanks for the suggestion. We agree that human activities such as groundwater pumping has an effect on the prediction of groundwater level. Since we do not have detailed pumping data in this area, the potential effect has been added in discussion 4.4. Meanwhile, we are trying to add electricity consumption data in the model, which is a factor to reflect the amount of groundwater pumping indirectly. This will be discussed in our future work.